# TripleThreat: Benchmarking Functional Sensitivity in Protein Representations with Paralog-Ortholog Triplets

**Mohini K. Misra**
Department of Bioengineering
Stanford University
mmisra@stanford.edu

**Gowri Nayar**
Department of Biomedical Data Science
Stanford University
gnayar@stanford.edu

**Emma Lundberg**
Departments of Bioengineering and
Pathology
Stanford University

**Russ B. Altman**
Departments of Bioengineering, Biomedical
Data Science, Genetics, Medicine, and
Computer Science
Stanford University

## Abstract

Understanding why certain sequence changes lead to functional change while others conserve function remains a central challenge in protein biology. A meaningful protein representation should be sensitive to these distinctions, so we introduce TripleThreat, a benchmark dataset to evaluate this capability. We construct test cases using natural examples of divergence and conservation in paralogs and orthologs, respectively, assembling protein–paralog–ortholog triplets. By controlling for confounders (sequence identity, length, and species) at varying levels, we generate six dataset subsets that trade off dataset size and stringency. Because protein language models (pLM) are popularly used as protein representations, we evaluate five widely used pLMs on this benchmark. We find that performance declines as confounding variables are more tightly controlled. Further, by varying confounder stringency, we identify which confounding signals overshadow functional signals; for example, an alignment-based pLM is observed to encode species identity more prominently than function. In sum, this work offers a framework to test whether protein representation spaces capture fine-grained functional relationships beyond confounding signals. We make our benchmark publicly available at https://github.com/mohinimisra26/triple-threat.

## 1 Introduction

Despite being essential for interpreting and engineering biology[1], the protein sequence–function landscape remains poorly understood. Sequence similarity is a strong correlate of function, but they are not causally linked; identical levels of similarity can yield either functional conservation or divergence. Evolution provides many examples of this: a protein's orthologs often transfer annotation across speciation events (a pattern formalized by the Ortholog Conjecture), whereas its paralogs frequently sub- or neo-functionalize after gene duplication, even at matched sequence similarity [2].

A biological representation that captures fine-grained functional differences should be able to distinguish between conserved orthologs and divergent paralogs. We introduce TripleThreat, a benchmark designed to assess this. We construct a dataset of protein triplets consisting of a reference protein, one of its paralogs, and one of its orthologs, all with matched sequence identity. This dataset is stratified into six subsets, each balancing dataset size and the stringency of confounder-removing parameters (including sequence length and species of origin) to varying degrees.

Most existing benchmarks evaluate single-protein properties, such as variant effects or functional annotation, without assessing relational semantics in representation space [3] [4]. Though multi-sequence tasks like protein–protein interaction prediction exist, they also do not systematically assess the quality of relational signals between representations. This benchmark fits into this gap.

We benchmark five protein language models (pLMs), which are a popular approach for protein representation learning. Trained on vast sequence data, pLMs produce embeddings that capture evolutionary constraints and context, and they are widely used for downstream biological tasks such as structure prediction and functional annotation[5]. However, most evaluations of pLMs assess performance using predictive models trained on top of these embeddings, leaving it unclear what biological meaning is encoded in the geometry of the embedding space itself [6]. This is important because many applications, such as understanding the function of designed proteins through nearest-neighbor search, would benefit from a representation that is organized with respect to function [7]. We therefore specifically benchmark here whether pLM embeddings *geometrically* distinguish between related paralogs and orthologs to evaluate functional sensitivty of representation space.

Our results show that, while several models distinguish paralogs from orthologs on less stringent subsets of the data, separability decreases as the confounding variables are controlled more tightly. The different performance between subsets also serves as a diagnostic tool, revealing which signals models rely on to distinguish proteins. For example, we find that alignment-reconstruction models appear to prioritize phylogenetic relationships over functional relationships. Taken together, TripleThreat reveals a lack of evolutionary functional specificity in current pLMs, and it is offered as a general-purpose benchmark framework to guide future functional representation learning.

## 2 METHODS

### 2.1 CONSTRUCTING A BENCHMARK TO ASSESS FUNCTIONAL SENSITIVITY

We construct a benchmark dataset of triplets consisting of a reference protein (A), a functionally divergent paralog (B), and a functionally conserved ortholog (C). This yields two matched pairs: {A-B} (divergent) and {A-C} (conserved).

To build this dataset, we use vertebrae-subset evolutionary trees from PANTHER [8] and identify duplication nodes. Since some paralogs are redundant in function, we run DIVERGE statistical analysis [9] on all duplication nodes and filter for those that provide evidence for functional divergence (83% of nodes pass). For each of these nodes, proteins descending from different children (i.e. they are in different clades) are considered paralogs $\{A\text{-}B\}$. For each $\{A\text{-}B\}$ pair, we calculate the pairwise sequence identity, $s$. Selecting for one member ($A$), we search for an ortholog ($C$) in the same clade that shares the aforementioned sequence identity, $s$, within a 1% threshold. This removes evolutionary distance as a confounder across the paralog-ortholog triplet. To ensure the resulting ortholog pair ($A\text{-}C$) did not experience functional divergence, we require that no duplication node lies on the shortest path between the two ortholog members in the evolutionary tree (Fig.1A).

To further ensure observed differences between paralogs and orthologs reflect functional divergence rather than sequence length or species effects, we enforce tunable constraints on both factors. (This produces multiple datasets, from strictly controlled to more relaxed, allowing users to trade off stringency for larger training sets that could be used downstream in learning tasks or statistical analyses with more power.) First, for sequence length, pair members are either required to be identical in length or, in a more lenient setting, all members of a triplet must be within 5% length of the shortest member. Second, we specify the combinations of species allowed to be in each pair. Because orthologs must come from different species, we stratify triplets as follows: (1) the paralogs (A–B) are from the same species, (2) the paralogs are from different species but with the ortholog (C) matching the species of paralog B, or (3) there is no species overlap within the triplet. The subsets are independent with respect to the species constraint, but nested with respect to the length constraint. We list them below, ordered from least to most stringent, followed by the number of triplets in each.

1. Unmatched paralog species, lenient sequence length match (32,931) *(Least stringent)*
2. Unmatched paralog species with ortholog constraint, lenient sequence length match (6,377)
3. Matched paralog species, lenient sequence length match (3,750)
4. Unmatched paralog species, exact sequence length match (1,461)
5. Unmatched paralog species with ortholog constraint, exact sequence length match (411)
6. Matched paralog species, exact sequence length match (191) *(Most stringent)*

Intuition for which signals each subset targets is elaborated on in A.1. The full benchmark dataset includes 42,239 triplets, with 17,127 proteins from 18 species and 981 protein families. We only

consider triplets with more than 20% sequence identity $s$, and in Fig.1B we show the number in each subset per sequence identity bin.

## 2.2 PROTEIN LANGUAGE MODEL REPRESENTATIONS USED FOR EVALUATION

We evaluate five diverse pLMs on this benchmark. The models span architectures and training data: ESM-2[10] and ProtT5[11], trained with masked language modeling on single-sequence corpora; ESM-C[12], a larger-scale ESM variant; ESM-1v[13], optimized for variant-effect prediction; and MSA Transformer[14], trained on multiple sequence alignments to model evolutionary correlations. We extract embeddings for all proteins in the paralog–ortholog dataset (Section 2.1) for each model.

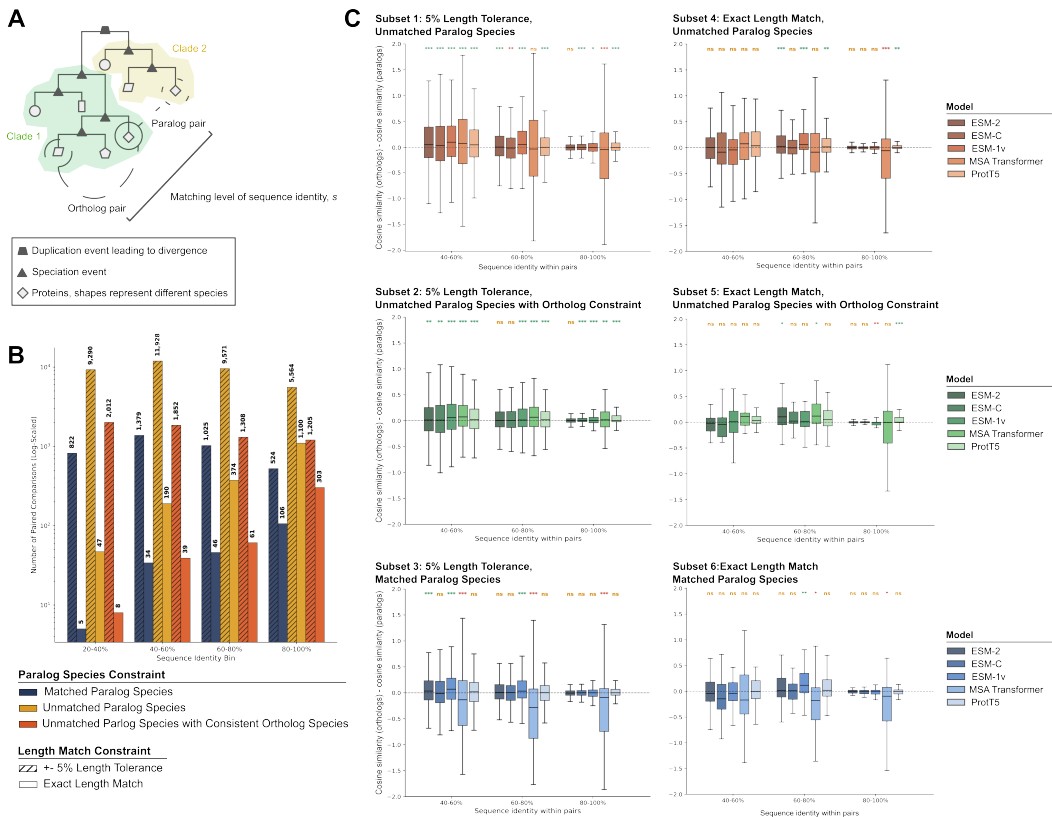

Figure 1: **(A)** Schematic of ortholog–paralog triplet construction from an evolutionary tree. Divergence at a duplicaiton node creates distinct clades that later undergo speciation. Paralogs are drawn from different clades (here, from the same species), while orthologs come from the same clade but different species. Orthologs are further restricted to share the most recent duplication ancestor. Triplets are selected so paralog and ortholog pairs have matched sequence identity $s$. **(B)** Composition of six benchmark subsets across sequence identity bins (20–40%, 40–60%, 60–80%, 80–100%). Subsets represent all permutations of three paralog species constraints and two length match constraints (see figure key for details). Y-axis is log-scaled. **(C)** The difference in cosine similarity of orthologs and paralogs across subsets. Values are plotted for 5 pLMs (ESM-2, ESM-C, ESM-1v, MSA Transformer, ProtT5) across sequence pair identity bins. Significance is calculated using a 2-sided Wilcoxon signed-rank test, indicated as * ($p < \frac{0.05}{n}$), ** ($p < \frac{0.01}{n}$), and *** ($p < \frac{0.001}{n}$), n = number of bins tested for each model. *ns* values are in yellow, significance indicating that orthologs have higher similarity than paralogs is in green, and significance indicating that paralogs are more similar than orthologs is in red. As confounders are removed, ability to discern paralogs from orthologs deteriorates.

## 2.3 EVALUATING A REPRESENTATION SPACE'S ABILITY TO SEPARATE FINE-GRAINED FUNCTIONAL SIGNALS

We compare paralog–ortholog pairs within triplets to provide a controlled test of whether they separate differently in representation space. Because pLM embeddings reflect global variation across protein families, within-family distances are small, so we center embeddings by subtracting their

PANTHER-defined family centroid to emphasize local functional differences (note, all triplet members are drawn from the same family). Triplets are stratified into four pairwise sequence identity ($s$) bins (20–40%, 40–60%, 60–80%, 80–100%). Within each bin, we evaluate the differences between paralog and ortholog similarities using two-sided Wilcoxon signed-rank tests[15] with Bonferroni correction[16]. We do so using both cosine similarity and Euclidean distance as a metric.

We also calculated the accuracy of each model across benchmark subsets and sequence identity bins. Accuracy is the percent of triplets where the paralog pair is less similar than the ortholog pair, capturing directional correctness rather than statistical significance relative to the overall distribution.

## 3 RESULTS

### 3.1 EMBEDDING SEPARABILITY DECREASES WITH STRINGENT CONFOUNDER CONTROLS AND HIGHER SEQUENCE SIMILARITY.

We present results for all benchmark subsets in Fig. 1C. Pairwise sequence identities $s$ below 40% are excluded due to limited data in the most stringent subsets (5 and 6; 20–40% bin: $n = 8$ and 5 triplets; Fig. 1B). Boxplots show the cosine-similarity difference between matched ortholog and paralog pairs. Significance is marked as * ($p < 0.017$), ** ($p < 0.0033$), and *** ($p < 0.00033$), post-Bonferroni adjustment; positive effects (orthologs more similar) are green, negative effects red, and non-significant results orange.

Without controlling for species or sequence length (Subsets 1 and 2, Fig. 1C, top left and middle left), pLMs generally distinguish paralogs from orthologs at low sequence identity $s$ (40–60%), though some become inconsistent at higher identity. For instance, in Subset 1 (Fig. 1C, top left), ESM-C places paralogs closer than orthologs in the 60–80% bin ($p = 0.0029$) but reverts to the opposite separation pattern in the 80–100% bin.

ProtT5 and ESM-1v are models that most consistently have positive effects on Subsets 1 and 2, separating paralogs more than orthologs across bins ($p < 0.017$). However, this advantage weakens—across all models—when paralogs are constrained to the same species or to exact length matches (Subsets 3-6), and no model shows positive significant effects across all identity ranges and conditions. The same is true when measuring for Euclidean distance (A.3). Overall, this implies that functional signals in these models are entangled with sequence similarity, species, and length effects.

Though some models display significantly positive effects, the median accuracy across all is 53.69%, and on the most stringently controlled dataset (Subset 6), no tested pLM has consistent performance above 50% (see A.2). Note that accuracy of 50% implies random performance.

To ground these results in biological implications, consider the following triplet identified by Uniprot IDs: { Protein: Q9QZL0, Ortholog: W5M6P5, Paralog: A0A3B3HFJ1 }. The protein and ortholog share an Enzyme Commission code while the paralog does not (2.7.11.1 v.s 2.7.11.25 respectively). However, all 5 pLMs display negative effects on this triplet; a paralog that is explicitly annotated to be functionally divergent is not organized as so in any of the tested pLM representation spaces.

### 3.2 DIFFERENT BENCHMARK SUBSETS REVEAL REPRESENTATION SPACE PRIORITIES

In Subset 1 (Fig. 1C, top left), where species membership is uncontrolled, MSA Transformer does not reliably separate paralogs from orthologs and, at the highest sequence similarity, even rates paralogs as more similar than orthologs ($p < 0.00033$, negative direction; red stars). When we instead match paralogs and orthologs by species (Subset 2, Fig. 1C, middle left), this pattern reverses (positive effects; green stars). This suggests that species differences can dominate the functional signal for MSA Transformer, since positive effects emerge only after controlling for species. To reinforce this hypothesis, when paralogs are forced to share species (Subsets 3 and 6, Fig. 1C, bottom), MSA Transformer often assigns them higher similarity than orthologs ($p < 0.00033$ in Subset 3; $p < 0.017$ or *ns* in Subset 6).

## 4 DISCUSSION

We introduce TripleThreat, a benchmark with subsets that progressively control confounders to test whether protein representations separate functional differences. Across widely used pLMs, none consistently distinguish functionally divergent paralogs from functionally conserved orthologs, providing limited evidence that these embedding geometries encode functional semantics with high sensitivity. Instead, signals such as species can dominate, as seen with MSA Transformer.

Poor performance has concrete implications for how pLMs are used in practice. If functionally divergent paralogs can be embedded more closely to each other than proteins that share annotation (as shown in the EC annotation example), then nearest-neighbor approaches to functional annotation risk incorrect labels, potentially undermining a widely used paradigm for protein function prediction.

A limitation to this study is our reliance on DIVERGE to infer functional divergence. Though perhaps imperfect, it offers a scalable way to annotate pairs across evolutionary trees without manual curation biases because it uses statistical approaches.

Although evaluated here on pLMs, TripleThreat is model-agnostic and extends to e.g. structure-based or multimodal methods, providing a general framework to test whether protein embeddings capture functional relationships rather than incidental evolutionary similarity.

### MEANINGFULNESS STATEMENT

A meaningful representation of life should not only capture statistical patterns in biological data, but also uncover higher-order relationships that remain inaccessible to traditional approaches. Because the underlying logic of many biological processes is still poorly understood, such representations should help illuminate these principles.

Our work contributes to this goal by providing a framework to directly test whether protein representation spaces encode higher-order functional relationships or are driven primarily by other signals. In doing so, this benchmark sets a higher bar for evaluating protein representations, guiding the development of more informative representation learning methods.

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

# A  APPENDIX

## A.1  INTUITION FOR BENCHMARK SUBSETS

Here we provide details about which benchmark subsets are suited for different investigations of representation space relationships.

1. *Unmatched paralog species, lenient sequence length match (32,931)-* This is the largest data set with the least-stringent confounder-control. Performing well on this task would indicate that a model can leverage weak sequence length signals to separate paralogs from orthologs. Because species is not constrained, sequence drift is an uncontrolled factor; paralog pairs could be separated by species-based differences, but so could orthologs, and the distribution of drift magnitude is not annoatated. Comparing results on this subset to results on a species-constrained subset could provide insight into the role of species in representation space organization. Also, since this is a large data set, it could be used for downstream model training.

2. *Unmatched paralog species with ortholog constraint, lenient sequence length match (6,377)-* Performing well on this task would also indicate that a model can leverage weak sequence length signals to separate paralogs from orthologs. Since {A, B} experiences the same species difference as {B, C}, this dataset evaluates if species drift can be distinguished from divergence + drift.

3. *Matched paralog species, lenient sequence length match (3,750)* - As above, performing well on this task would indicate that a model can leverage weak sequence length signals to separate paralogs from orthologs. Since {A, B} experiences no species drift, this dataset evaluates if species drift can be distinguished from divergence.

4. *Unmatched paralog species, exact sequence length match (1,461)-* Performing well on this task would indicate that a model can separate paralogs from orthologs independent of sequence length. Because species is not constrained, sequence drift is an uncontrolled factor; paralog pairs could be separated by species-based differences, but so could orthologs, and the distribution of drift magnitude is not annoatated. Comparing results on this subset to results on a species-constrained subset could provide insight into the role of species in representation space organization.

5. *Unmatched paralog species with ortholog constraint, exact sequence length match (411)-* Performing well on this task would indicate that a model can separate paralogs from orthologs independent of sequence length. Since {A, B} experiences the same species difference as {B, C}, this dataset evaluates if species drift can be distinguished from divergence + drift.

6. *Matched paralog species, exact sequence length match (191)-* Performing well on this task would indicate that a model can separate paralogs from orthologs independent of sequence length. Since {A, B} experiences no species drift, this dataset evaluates if species drift can be distinguished from divergence.

Beyond being a simple gradient of confounders, these datasets can be used to answer different questions about what a model can detect. Additionally, they can be contrasted or combined as the user pleases to extract the information desired.

## A.2 ACCURACY RANKING OF pLMs ON THE BENCHMARK

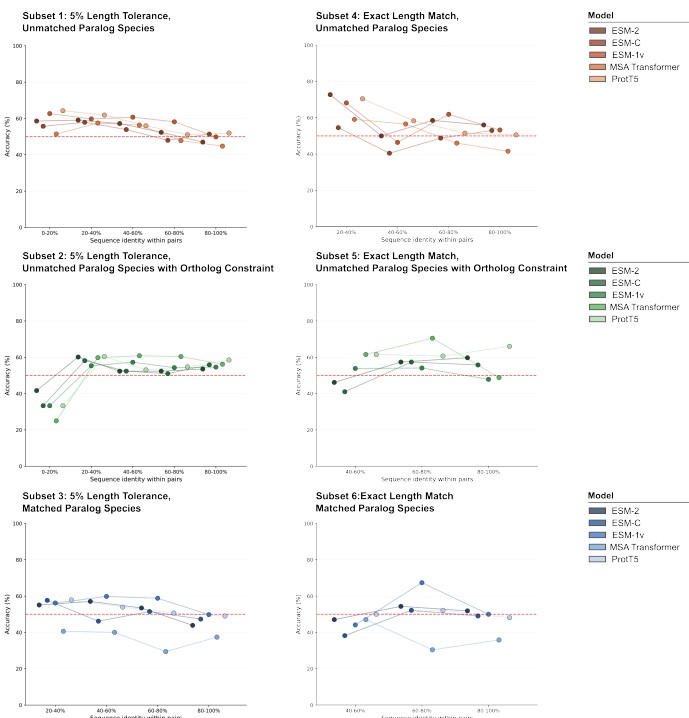

Figure 2: Accuracy results across the tested pLMs for the 6 benchmark subsets, divided by sequence identity bin. Overall, these pLMs do not reliably have orthologs as more similar than paralogs.

## A.3 Results across all benchmark tasks, Euclidean distance metric

N/A values are due to there not being enough data points for statistical evaluation.

**ESM-1v**

| | Percent Identity Bin | | | |
|---|---|---|---|---|
| Benchmark Type | 20-40% | 40-60% | 60-80% | 80-100% |
| 1 | *** | ns | *** | ns |
| 2 | *** | * | * | *** |
| 3 | * | ns | ns | ns |
| 4 | ns | * | ns | ns |
| 5 | N/A | ns | ns | ns |
| 6 | N/A | ns | ns | ns |

**ESM-2**

| | Percent Identity Bin | | | |
|---|---|---|---|---|
| Benchmark Type | 20-40% | 40-60% | 60-80% | 80-100% |
| 1 | *** | ns | *** | *** |
| 2 | *** | ns | ns | ns |
| 3 | *** | ns | *** | *** |
| 4 | ns | ns | ns | ns |
| 5 | N/A | ns | ns | ns |
| 6 | N/A | ns | ns | ns |

**ESM-C**

| | Percent Identity Bin | | | |
|---|---|---|---|---|
| Benchmark Type | 20-40% | 40-60% | 60-80% | 80-100% |
| 1 | *** | *** | *** | * |
| 2 | *** | ns | ns | *** |
| 3 | ** | *** | * | ns |
| 4 | ns | * | ns | ns |
| 5 | N/A | ns | ns | ** |
| 6 | N/A | ns | ns | ns |

**MSA Transformer:**

| | Percent Identity Bin | | | |
|---|---|---|---|---|
| Benchmark Type | 20-40% | 40-60% | 60-80% | 80-100% |
| 1 | *** | *** | *** | *** |
| 2 | *** | *** | *** | ** |
| 3 | *** | *** | *** | *** |
| 4 | ns | ns | * | *** |
| 5 | N/A | ns | * | ns |
| 6 | N/A | ns | ** | * |

**ProtT5:**

| Benchmark Type | Percent Identity Bin | | | |
|:---:|:---:|:---:|:---:|:---:|
| | 20-40% | 40-60% | 60-80% | 80-100% |
| 1 | *** | *** | *** | ** |
| 2 | ns | *** | ns | *** |
| 3 | ns | *** | *** | ns |
| 4 | * | ns | ns | *** |
| 5 | N/A | * | ns | *** |
| 6 | N/A | ns | ns | ns |

