# OpenReview forum: "TripleThreat: Benchmarking Functional Sensitivity in Protein Representations with Paralog-Ortholog Triplets"
_ICLR.cc/2026/Workshop/LMRL — ICLR 2026 Workshop LMRL Poster_

### Official Review · Reviewer_jXZe · 2026-02-23
**Using ortholog vs. paralogs to assess representations of protein function**

**Rating:** 8
**Confidence:** 4

**Review:**

This paper introduces a challenging protein function benchmark for protein language models. Rather than relying on functional data or measurements, the authors take advantage of the widely accepted idea that orthologs are usually closer in function than paralogs, even if the have similar levels of sequence similarity. The authors argue that protein language embeddings should put the orthologs closer together than paralogs and their success at this task is a measure of the degree to which language models represent protein function.

I think this is a great line of thinking and that this kind of clever use of evolutionary principles is a valuable way forward for assessing these models. However, I think the paper could be improved in terms of clarity, especially in motivating their new task and also in explaning the 6 levels of difficulty in their benchmark. Machine Learning researchers training protein language models may not appreciate the 'ortholog hypothesis' or the nuance of controlling for protein length and sequence identify.

For example, the authors write: " Sequence similarity alone is an unreliable
proxy for function: in some cases, proteins with comparable similarity retain annotation, while in
others, function diverges"

I get what the authors are trying to say here, but I don't think they are conveying the needed nuiance. After all, BLAST, Pfam, Prosite, etc., are all very good proxies for function and they are all based on sequence similarity. The authors point is that *despite* being an excellent proxy for function, sequence similarity is only a strong *correlate* of function, not a casual determinant. This is the real motivation their work. Can state-of-the-art methods still predict function when sequence similarity is controlled for?

The authors also should clarify that orthology and paralogy don't guarantee functional conservation vs. change, but they again, they are correlated with it.

---

### Official Review · Reviewer_RQwf · 2026-02-25
**Good proposal work on benchmarking dominant embedding geometry**

**Rating:** 7
**Confidence:** 4

**Review:**

Summary: This paper proposes TripleThreat, a benchmark built from ortholog–paralog triplets designed to test whether protein embeddings reflect functional divergence rather than shortcuts like sequence identity, species, or length.

**Strengths:**
- Good benchmark idea. Matched-identity triplets + a strictness ladder is a clean way to test what signals dominate embedding geometry.
- Reasonable diagnostic with a subset progression that surfaces interpretable failure modes

**Weaknesses:**
- The use of proxy by DIVERGENCE is the weakest point since its not validated with functional annotations.
- Evaluation is not that straightforward since it's done via pvalues. It would be great to have a ranking by acc.
- Sample size in subsets is too small to derive statistically significant conclusions.

I recommend the acceptance acknowledging that it is within the tiny paper track of a workshop.

---

### Official Review · Reviewer_xNrq · 2026-02-26

**Rating:** 5
**Confidence:** 5

**Review:**

The paper introduces TripleThreat, a benchmark intended to probe whether protein language models capture functional divergence by distinguishing orthologs from paralogs in curated triplets. The authors define six benchmark variants that manipulate sequence-identity regimes and species composition to mitigate confounders, and evaluate four protein language models along with MSA Transformer. Across settings, performance is low and appears strongly confounded by species, suggesting current representations may emphasize phylogenetic/taxonomic structure over the functional distinctions targeted by this task.

I find the benchmark design interesting and appreciate the effort to control confounders via identity matching and species-balanced variants. However, I think the evaluation still misses an important source of bias: pretraining data exposure. Differences in model performance could reflect how heavily a family/taxon is represented in the pretraining corpus, and identity matching alone does not control for this. The paper would be strengthened by stratifying results using an exposure proxy (e.g. nearest-neighbor counts in UniRef) and seeing if this could explain the lack of performance.

A second concern is external validity: many prior works report strong results for PLMs on function-prediction benchmarks, whereas TripleThreat highlights a failure on ortholog–paralog discrimination. Biologically these are related, but the paper does not yet make a clear computational link between this diagnostic and downstream functional performance. I would like to see an analysis connecting TripleThreat outcomes to a practical task. For example, whether proteins/families where a model fails on TripleThreat also show poorer performance on function annotation transfer or related functional benchmarks. Making this connection explicit would clarify why this benchmark matters beyond its biological motivation.

Lastly, I am not fully convinced that ortholog–paralog discrimination is the right (or uniquely informative) proxy for “functional sensitivity” as framed here. There are many established PLM evaluations for functional and evolutionary signal, but the paper does not position TripleThreat relative to these or demonstrate what failure mode it uniquely reveals. It would strengthen the work to either (i) explicitly argue why this benchmark captures a distinct capability not measured by prior benchmarks, or (ii) empirically relate TripleThreat scores to performance on one or two standard function-focused benchmarks under comparable similarity-controlled splits. Without that comparative grounding, it is hard to assess what new insight TripleThreat provides beyond the biological motivation.

It is for these reasons I give the paper a weak reject.

---

### Meta-Review · Area_Chair_2y1M · 2026-02-27

**Recommendation:** Accept (Poster)
**Confidence:** 4

**Metareview:**

Accept.

---

### Decision · Program_Chairs · 2026-03-02

**Decision:**

Accept (Poster)

**Comment:**

Please see the meta-review.